# Using Machine Learning and Generative Intelligence in Book Cover Development

**DOI:** 10.3390/jimaging11020046

**Published:** 2025-02-07

**Authors:** Nonna Kulishova, Daiva Sajek

**Affiliations:** Kauno Kolegija Higher Education Institution, Faculty of Informatics, Engineering and Technologies, Pramonés av. 20, 111965284 Kaunas, Lithuania; daiva.sajek@go.kauko.lt

**Keywords:** generative artificial intelligence, machine learning, k-means, clustering, color, book covers, self-publishing

## Abstract

The rapid development of machine learning and artificial intelligence approaches is finding ever wider application in various areas of life. This paper considers the problem of improving editorial and publishing processes, namely self-publishing, when designing book covers using machine learning and generative artificial intelligence (GAI) methods. When choosing a book, readers often have certain expectations regarding the design of the publication, including the color of the cover. These expectations can be called color preferences, and they can depend on the genre of the book, its target audience, and even personal associations. Cultural context can also influence color choice, as certain colors can symbolize different emotions or moods in different cultures. Cluster analysis of book cover images of the same genre allows us to identify color preferences inherent in the genre, which is proposed to be used when designing new covers. The capabilities of generative services for creating and improving cover designs are also investigated. An improved flow chart for using GAI in creating book covers in the process of self-publishing is proposed, which includes new stages, namely exploring, conditioning, and evolving. At these stages, the designer creates prompts for GAI and examines how they and GAI’s issuances correspond to the task. Conditioning allows for even more precise adjustment of prompts to features of each book, and the evolving stage also includes post-processing of results already received from GAI. Post-processing, in turn, can be performed both in generative services and by a designer. The experiment allowed us to use the machine-learning method to determine which colors are most often found in book cover layouts of one of the genres and to check whether these colors correspond to harmonious color palettes. In accordance with the proposed scheme of the design process using generative artificial intelligence, versions of book cover layouts of a given genre were obtained.

## 1. Introduction

The book industry in 2024 continues to thrive, driven by diverse consumer preferences, digital innovation, and global accessibility. The global book market was valued at approximately USD 144.67 billion in 2023, and it is expected to grow at a compound annual growth rate, according to various estimates, from 1.8 to 3% from 2024 to 2030 [1,2,3]. North America remains the largest regional market, contributing over 33% of global revenues in 2023. Print books continue to dominate, holding over 78% of the global revenue share in 2023. Despite the rise of digital formats, the tactile appeal and durability of print books maintain their popularity [1,2,3].

The role of book cover design has become a vital factor in determining a book’s commercial success. Covers act as the first point of contact with potential readers, influencing their decision to explore, purchase, or bypass a book entirely [4,5,6]. Book cover design is no longer just a decorative feature or an afterthought in the book production process. It has become a vital part of marketing, storytelling, and reader engagement. Whether for printed books or digital versions, the importance of book cover design continues to grow, reflecting changing trends in technology, art, and consumer behavior. Nowadays, generative artificial intelligence (GAI) tools are actively included in the design creation process. A 2024 report from Design Intelligence reveals that 60% of publishers have incorporated AI tools into various stages of book production, with 35% using them specifically for cover design. In addition, a survey by BookTech showed that 70% of self-published authors experimented with AI-powered design tools in 2023, a significant increase from 45% in 2020 [7].

Book cover design has come a long way from its initial purpose as a protective layer for bound pages. Early books were often wrapped in plain leather or cloth. Over time, as books became more accessible and mass-produced, their covers began to play a dual role: protecting the contents and communicating information about the book. During the medieval period, books were bound in intricate materials like gold-leafed leather or ivory, making them precious objects of art. These designs were not about marketing but about prestige. The invention of the printing press in the 15th century revolutionized book production. Covers began to include basic information like the title and the author. As literacy rates grew, publishers realized the potential of covers to attract readers. This marked the beginning of cover design as a tool for marketing. In the 20th century, book cover design became an established profession. Movements such as Art Deco, Modernism, and Pop Art influenced the aesthetics of covers. Designers started experimenting with bold colors, abstract forms, and typography to create iconic covers that remain memorable to this day. By the late 20th century, computers allowed for digital design, introducing even greater possibilities for creativity.

As the publishing industry embraces digital platforms, book covers must adapt to new challenges and opportunities. E-books and online marketplaces require different design considerations compared with physical books. In digital marketplaces, covers are often displayed as thumbnails. This means they must be legible and appealing at small sizes. Key elements like the title and the author name must remain clear.

Online platforms rely on algorithms to promote books. A cover design that aligns with a book’s metadata (keywords, genre tags, etc.) can improve its visibility. Social media platforms have become important for book marketing. A visually stunning cover can go viral, boosting sales through shares and hashtags. Many authors and publishers now incorporate cover reveal campaigns to build excitement.

Notwithstanding the advancements in tools and techniques, book cover design faces several challenges: balancing creativity and marketability, catering to diverse audiences, budget constraints, and staying relevant. Designers must create covers that are both visually appealing and commercially viable. Too much creativity can alienate readers, while overly commercial designs may feel generic. Books are sold globally, so covers must appeal to a broad audience while respecting cultural sensitivities. Colors, symbols, and imagery must be carefully chosen to avoid unintended offense. Independent authors and small publishers often have limited budgets, which can restrict the quality of the cover design. However, investing in a professional designer usually pays off in terms of sales and reader perception. Trends in design are constantly evolving. Designers must strike a balance between staying current and creating timeless designs that will not feel dated in a few years.

AI-powered platforms such as Canva [8], Adobe Sensei [9], and Book Brush [10] allow users to create professional-looking book covers with minimal design expertise. These tools offer templates, automated suggestions for color schemes, font pairings, and layout adjustments, reducing the time and effort required to produce visually appealing designs. Additionally, AI tools analyze genre-specific trends to recommend elements that resonate with target readers, helping self-published authors craft market-relevant covers [11].

Despite its benefits, using AI in book cover design has limitations. AI tools may lack the nuanced understanding of storytelling or genre-specific subtleties that human designers bring to their work. Over-reliance on templates and automated suggestions can result in generic or repetitive designs. Furthermore, ethical concerns about the originality of AI-generated content and its potential reliance on copyrighted material require careful consideration.

AI enriches book cover design by offering speed, affordability, and creative flexibility. However, balancing AI’s capabilities with human expertise ensures that covers remain unique, engaging, and true to the author’s narrative vision. This paper examines the problem of modifying the process of book cover design development using machine learning and generative artificial intelligence (GAI).

## 2. Background

The purpose of a cover extends beyond protecting the book. It serves multiple roles, from conveying the book’s theme and genre to persuading readers to pick it up. A book cover is both an art form and a marketing tool, requiring careful consideration of visual and psychological factors [12,13].

The traditional editorial and publishing process of preparing books for printing is carried out by qualified professionals using the appropriate software. In this process, the development of the cover design is an integral part of creating the style of the entire book, which also includes the tasks of choosing fonts, designing the layout, and creating illustrations. A harmonious combination of these elements makes the book more attractive to readers and, accordingly, increases its sales [14,15].

But now, along with traditional publishing processes, where authors bring a manuscript to a publishing house and designers do the rest, the process of self-publishing has become widespread, where an author can independently select illustrations, develop a layout and a cover for a book published in small print using digital printing, or distribute the book in electronic format [16]. The author of a book is not a professional in designing, so it is necessary to simplify the process of book layout developing so that even a non-professional can create a quality product. Numerous recommendations for preparing a book for printing suggest among the main steps the following:-Understand the audience and research the market.-Choose a distinct style, suitable to book genre.-Set the appropriate color.-Use high-quality illustrations [17].

These recommendations, upon closer examination, turn out to be interconnected: a particular book genre has its own audience with already formed expectations and ideas about what artistic style, color scheme, and visuals best suit the book. Book cover design is influenced by cultural trends, technological advancements, and evolving consumer preferences. Some of the most popular trends today include the following [18]:Rise of AINostalgia—50 s, 60 s, 70 s, etc. stylizationAbstract artMixing digital art and hand drawingsUsing font as visual elementsMinimalism with a twistMovie influenceCollagesBotanic styleActivistic themesDouble exposureTitle dominatedHeadshots

The rise of AI in book cover design has significantly transformed the creative process, allowing designers to generate and experiment with endless visual concepts quickly and efficiently, thus opening doors to more experimental and diverse styles. Nostalgic trends referencing the 50 s, 60 s, and 70 s often tap into cultural cycles of revival, where AI tools can quickly emulate and remix past design aesthetics, connecting contemporary readers with vintage visual cues. The increasing popularity of abstract art and mixed digital and hand-drawn styles reflects a desire for both innovation and authenticity, where AI allows seamless blending of different techniques and a more accessible way to merge traditional and modern forms. Minimalism with a twist and activism-themed designs are shaped by global social movements and the digital age, where simplicity and bold statements can be emphasized through AI-generated variations or optimized visual messaging that resonates with modern sensibilities. Trends like collages, double exposure, and movie-inspired covers benefit from the flexibility of AI, which can manipulate images and fonts in innovative ways, helping to tell visually complex stories that align with the growing need for immersive, visually dynamic book marketing.

All these current design trends are highly dependent on strategic color. Readers often subconsciously interpret colors based on their cultural or emotional significance [19,20,21,22,23]. Investigating how a target audience perceives specific color clusters ensures that the cover design connects with their expectations [24,25]. Without investigating the color content, designs risk appearing disconnected from the book’s essence or failing to stand out in a competitive marketplace. Digital considerations also make color investigation crucial. In the digital marketplace, covers must be optimized for thumbnails and various screen resolutions. Investigating color visibility and legibility ensures that a cover stands out even at small sizes. High-contrast colors are particularly important in digital platforms, where the competition for attention is fierce.

## 3. Methodology

There is currently a significant growth in the volume of publication of certain genres of printed books, which increases competition between individual publications and forces graphic designers to look for design solutions that will not allow the book to get lost in thousands of similar publications.

The purpose of this study is to improve the quality of book cover design by creating a harmonious color content of images. To achieve this goal, it is necessary to

-Develop a methodology for determining which colors predominate in the developed cover design layout.-Assess whether the color scheme of the design is harmonious.-If the colors selected for the cover layout are not harmonious, study the capabilities of generative services based on artificial intelligence for correcting and harmonizing colors in the layout.

The sequence of solving these problems is shown in Figure 1.

To use new effective generative tools in the practice of designers, it is necessary to establish whether there are any stable patterns during the creation of projects. The first stage of this study is finding the connection between the subjective ideas of readers and artists about visual images that should correspond to the genre and content of the book [26,27]. To determine whether there are established semantic links between book genres, design styles, and color palettes of published book covers, various methods of machine learning and artificial intelligence are used [28,29,30]. In this paper, it is proposed to use a well-known image clustering method called k-means [31].

K-means is often considered better than DBSCAN and hierarchical clustering for large datasets because it is computationally more efficient, with a time complexity of O(nk), where n is the number of data points and k is the number of clusters. Hierarchical clustering forms a multilevel data architecture that can be scaled. K-means is limited to one level of data partitioning and typically performs better when the clusters are roughly spherical and evenly sized, so it is much more effective in analyzing big data [28,29,30,32]. DBSCAN takes into account outliers in the data, grouping them into a separate cluster. This can be useful for detecting data outliers. But when it is unknown whether there will be outliers at all in the task, when generalized information about the presence of colors in a large array of images is more important, such features of DBSCAN seem redundant. In addition, DBSCAN is focused on forming clusters with uniform density; but with regard to color data from images, it cannot be guaranteed that they will actually fill the color space uniformly; rather, the opposite is true.

K-means clustering aims to partition a set of observations $x={\left(x_{1},x_{2},\ldots ,x_{n}\right)}^{T}$, where each observation is a *d*-dimensional real vector, into k (≤*n*) clusters ***S*** = {*S*_1_, *S*_2_, …, *S_k_*} to minimize the within-cluster sum of squares or variance. The objective is to find the values of *S_i_* that minimize the equation(1)$$\arg\underset{S}{\min}\sum_{i=1}^{k}\sum_{x\in S_{i}}{\left\|x-\mu_{i}\right\|}^{2}=\arg\underset{S}{\min}\sum_{i=1}^{k}\left|S_{i}\right|VarS_{i}$$
where $\mu_{i}=\frac{1}{\left|S_{i}\right|}\sum_{x\in S_{i}}x$ is the centroid or mean of the data points in *S_i_*, $\left|S_{i}\right|$ is the size of *S_i_*, and $\left\|\cdot \right\|$ is the L2 norm. This is equivalent to minimizing the pairwise squared deviations of points within the same cluster:$$\arg\underset{S}{\min}\sum_{i=1}^{k}\sum_{x,y\in S_{i}}{\left\|x-y\right\|}^{2}$$

As observations $x={\left(x_{1},x_{2},\ldots ,x_{n}\right)}^{T}$ in this work, CIE L*a*b* [33] color coordinates are accepted. CIE L*a*b* is a color space defined by the International Commission on Illumination (CIE) to clearly designate and measure colors that human can perceive. L* describes perceptual lightness; a* and b* are color hues in opposite scales: a* represents the red/green dimension, and b* represents the blue/yellow dimension. CIE L*a*b* represents a perceptual color space and has a wider gamut than other color spaces; it is a useful system for detecting small variations in color.

CIE L*a*b* is a device-independent color space, as opposed to, for example, RGB and CMYK. RGB (Red, Green, Blue) is an additive model of a device-dependent color space, commonly used for digital displays, where colors are created by combining varying intensities of red, green, and blue light [33]. CMYK (Cyan, Magenta, Yellow, Key/Black) is a subtractive color model used in color printing, where colors are produced by subtracting light using ink, and it works by mixing various percentages of cyan, magenta, yellow, and black inks [33]. This color space is device-dependent too. The main difference between these three color spaces lies in their purpose for consistent color reproduction across different devices and environments.

It is proposed to use cluster analysis of the color content of book covers to determine whether there are palettes that predominate in the design of books of certain genres. Color palette is a set of colors that a designer or artist chooses to work with on a project. It is convenient to create palettes using color wheels; the early wheels used RYB (Red, Yellow, Blue) primaries alongside secondary and tertiary colors [29,30,34,35]. Since designers use digital monitors that process images in the RGB model, color palettes are now more often influenced to use RGB (Red, Green, Blue) primaries (Figure 2).

An angular shift in the wheel corresponds to a change in hue, and a radial shift corresponds to a change in lightness. The entire color wheel is conditionally divided into 12 shades, so we can assume that each shade corresponds to 1/12 of the wheel, that is, a sector of 30 degrees.

Palettes are formed by radius vectors, each of which corresponds to one color. The most famous palettes are analogous, monochrome, complementary, triadic, and tetradic (Figure 3).

In color combinations for design, hue is more important than lightness. Thus, it is possible to introduce relations that describe the formation of one or another type of palette for two colors $C_{1}$ and $C_{2}$ using angle φ between correspondent radius vectors (Figure 4).

Analog palette condition:(2)$$\varphi \left(C_{1},C_{2}\right)=\frac{\pi }{6}+\delta$$

Monochromatic palette condition:(3)$$\varphi \left(C_{1},C_{2}\right)=0+\delta$$

Complementary palette condition:(4)$$\varphi \left(C_{1},C_{2}\right)=\pi +\delta$$

Complementary split palette condition:(5)$$\varphi \left(C_{1},C_{2}\right)=\pm \frac{5\pi }{6}+\delta$$

Triadic palette condition:(6)$$\varphi \left(C_{1},C_{2}\right)=\frac{2\pi }{3}+\delta$$

Tetradic palette condition:(7)$$\varphi \left(C_{1},C_{2}\right)=\frac{\pi }{4}+\delta$$
where $C_{1}$ is the vector from the color wheel center to the point with color coordinates $a_{1}^{*},b_{1}^{*}$; $C_{2}$ is the vector from the color wheel center to the point with color coordinates $a_{2}^{*},b_{2}^{*}$; δ is the angular tolerance, and in this work, it is accepted to be equal to $\frac{\pi }{12}$, a half from one color sector.

Color data from cover design after clustering are subject to verification for compliance with the conditions of belonging to a particular palette. Obviously, failure to comply with any of the conditions will indicate that the design does not look harmonious and should be changed.

To find which palette the color scheme of a particular image is closest to, the following approach is suggested. Among the seven most frequently occurring colors in an illustration, one will be dominant. Let us determine the cosines of the angles that the radius vectors of the other six colors form with this dominant color:(8)$$\underset{i=\bar{1,6}}{\cos}\left(\hat{C_{p},C_{i}}\right)=\frac{a_{p}^{*}a_{i}^{*}+b_{p}^{*}b_{i}^{*}}{\sqrt{{\left(a_{p}^{*}\right)}^{2}+{\left(a_{i}^{*}\right)}^{2}}\sqrt{{\left(b_{p}^{*}\right)}^{2}+{\left(b_{i}^{*}\right)}^{2}}}$$
where $C_{p}$ is the vector from the color wheel center to that predominated for this cover color with coordinates $a_{p}^{*},b_{p}^{*}$; $C_{i}$ is one of the six other vectors from the color wheel center to colors with coordinates $a_{i}^{*},b_{i}^{*}$.

For each image $I_{k}$, these cosines form a set(9)$$\begin{array}{l} D_{I_{k}}={\left.\left\{d_{I_{k}i}\right\}\right|}_{i=\bar{1,6}}=\\ =\left\{\cos\left(\hat{C_{p},C_{1}}\right),\cos\left(\hat{C_{p},C_{2}}\right),\cos\left(\hat{C_{p},C_{3}}\right),\cos\left(\hat{C_{p},C_{4}}\right),\cos\left(\hat{C_{p},C_{5}}\right),\cos\left(\hat{C_{p},C_{6}}\right)\right\} \end{array}$$

We will find how much these angles differ from the normative ones—π/6, 0, π, 2π/3, 5π/6, π/4—that describe the position of two vectors corresponding to a particular palette by conditions (1)–(6). To do this, we will sequentially subtract the cosine values of the normative angles from vector D:(10)$$\Delta_{I_{k}Ana\log}=\left|D_{I_{k}}-\cos\frac{\pi }{6}\right|$$(11)$$\Delta_{I_{k}Monochrom}=\left|D_{I_{k}}-\cos0\right|$$(12)$$\Delta_{I_{k}Compl}=\left|D_{I_{k}}-\cos\pi \right|$$(13)$$\Delta_{I_{k}Split}=\left|D_{I_{k}}-\cos\frac{5\pi }{6}\right|$$(14)$$\Delta_{I_{k}Triad}=\left|D_{I_{k}}-\cos\frac{2\pi }{3}\right|$$(15)$$\Delta_{I_{k}Tetrad}=\left|D_{I_{k}}-\cos\frac{\pi }{4}\right|$$

These differences add along every angle ${\left.\left\{d_{I_{k}i}\right\}\right|}_{i=\bar{1,6}}$ between the most frequently encountered color in the image:(16)$$Div_{I_{k}Ana\log}=\sum_{i=1}^{6}\Delta_{I_{k}Ana\log i}=\sum_{i=1}^{6}\left|d_{I_{k}i}-\cos\frac{\pi }{6}\right|$$(17)$$Div_{I_{k}Monochrom}=\sum_{i=1}^{6}\Delta_{I_{k}Monochrom\;i}=\sum_{i=1}^{6}\left|d_{I_{k}i}-\cos0\right|$$(18)$$Div_{I_{k}Compl}=\sum_{i=1}^{6}\Delta_{I_{k}Compl\;i}=\sum_{i=1}^{6}\left|d_{I_{k}i}-\cos\pi \right|$$(19)$$Div_{I_{k}Split}=\sum_{i=1}^{6}\Delta_{I_{k}Split\;i}=\sum_{i=1}^{6}\left|d_{I_{k}i}-\cos\frac{5\pi }{6}\right|$$(20)$$Div_{I_{k}Triad}=\sum_{i=1}^{6}\Delta_{I_{k}Triad\;i}=\sum_{i=1}^{6}\left|d_{I_{k}i}-\cos\frac{2\pi }{3}\right|$$(21)$$Div_{I_{k}Tetrad}=\sum_{i=1}^{6}\Delta_{I_{k}Tetrad\;i}=\sum_{i=1}^{6}\left|d_{I_{k}i}-\cos\frac{\pi }{4}\right|$$ and forms the final feature vector(22)$$\begin{array}{l} P_{I_{k}}={\left.\left\{p_{I_{k}i}\right\}\right|}_{i=\bar{1,6}}=\\ =\left\{Div_{I_{k}Ana\log},Div_{I_{k}Monochrom},Div_{I_{k}Compl},Div_{I_{k}Split},Div_{I_{k}Triad},Div_{I_{k}Tetrad}\right\} \end{array}$$

To decide if color correction of an image $I_{k}$ is necessary, we need to determine to what palette does it belong. To do this, we find the argument of the minimum of the vector $P_{I_{k}}$:(23)$$\begin{array}{l} PaletteType=\arg\underset{T}{\min}\left(P_{I_{k}}\right),\\ T=\left\{Ana\log,Monochrom,Compl,Split,Triad,Tetrad\right\} \end{array}$$

Changing the color scheme of the design is quite a difficult task. Its solution can be simplified if the design uses a small number of elements, all of which have a single coloring and clearly defined contours. In this case, any graphic editor (Adobe Photoshop v26.3, Adobe Illustrator v29.2.1, etc.) and its tools to replace the color of one or more elements can be useful. If the design contains gradients, photorealistic images, or photographs, then the gamut can be changed using the color correction tools of the same graphic editors. Such operations take more time and require high qualifications of the designer; they can only be performed by professionals; therefore, they are applicable only to traditional publishing processes.

For self-publishing, complex color correction may be unavailable, and the author of the book will most likely not be so qualified. Therefore, to harmonize the design, the author can be offered the use of generative artificial intelligence.

The swift progress of technologies leveraging artificial intelligence, particularly text-to-image generation, is transforming our approach to design across various fields. This includes game design [37,38,39], web design [40], and multimedia or print publications [41]. As a result of numerous similar studies, a distinct field known as generative artificial intelligence (GAI) has emerged. This method is undeniably appealing to professionals due to its virtually limitless creative potential, offering numerous options and speeding up the process of producing artistic concepts.

Generative adversarial networks (GANs) [42] were among the first solutions for the artificial creation of images containing objects and have now become widely popular in image generation. GANs consist of two main components: the generator and the discriminator. The generator learns to replicate the statistical distribution of real examples to generate new data, while the discriminator attempts to distinguish whether the input data are real or fake. The design of both the generator and the discriminator plays a critical role in the stability and effectiveness of GANs. Traditional convolutional GANs generate high-resolution details by relying solely on spatially localized points in lower-resolution feature maps, which makes generating diverse high-resolution samples from complex datasets a challenging task.

## 4. Results

### 4.1. Covers Color Content Analysis

To explore the possibility of identifying the most commonly used color palettes for books of a certain genre, a dataset with romance book covers [43] was used. This dataset was gathered within twelve years 2011–2023 and contains 1434 cover images, author names and titles, and brief book annotations. The images are photographs of book covers sold on various online services in the romance novel genre. The author of the dataset examined, for example, aspects such as the presence and the number of partially clothed or unclothed people on the cover, and whether the cover illustrations are more hand-drawn or photorealistic. In particular, the author provides a diagram of the use of images of partially or fully unclothed people on romantic novel covers. The diagram shows that over time, such images on covers are becoming fewer, which leads to a reduction in skin tone representation in design. Another diagram by the author demonstrates that in recent years, the number of photorealistic illustrations in the design of romance covers has significantly decreased, and the vast majority of covers are now hand-drawn.

To discover what colors were used on each cover, clustering using the k-means method was performed, and the 7 most frequently occurring colors were identified. The 10,038 colors collected in this way are shown in Figure 5 as dot volume. Every dot denotes one color in the CIE L*a*b* space.

A slight concentration of dots in the dark part of the diagram can be noted. In the rest of the volume, the dots are distributed fairly evenly; however, green colors are underrepresented. In this array of points, k-means clustering was also carried out, and 30 most frequently repeated colors were identified. These colors and their LAB coordinates are shown in Figure 6.

Despite the fact that in recent years, cover designs with images of fully or partially naked people have become less and less popular, this study found that, among the 30 most common colors on romance covers, 9 (that is, almost a third) can be classified as skin tones. These colors were compared with data from [44,45]. Six colors are shades of blue, 5 are red and violet, 4 are shades of neutral gray, 4 are shades of yellow-brown, and only 2 colors are green. So, the now discovered fact is that a third of the most common colors on romance covers are skin colors, despite the images of partially clothed people becoming fewer. It is clear from the clustering results that the skin color gamut in the perception of designers, authors, and readers is closely related to romantic novels’ content.

According to [46], red colors are associated with love, purple and pink with beauty and femininity, and blue shades with purity. In romance books, for example, love often emerges as a transformative force, bringing characters closer and allowing them to discover deeper connections. Beauty, femininity, and purity are frequently depicted through idealized characters or settings, where these traits symbolize innocence, emotional depth, and the potential for a perfect love story.

To investigate how the colors on covers are harmoniously combined and which palettes predominate, conditions (1)–(6) were tested for a set of the 7 most frequently occurring colors on each cover. This test showed that, given the accepted tolerance, 144 images have an analog palette, 117 images have a monochromatic palette, 12 images have a triadic palette, 11 images have a tetradic palette, and 27 images can be classified as those that have both analog and monochromatic palettes (Figure 7). At the same time, 1177 illustrations did not meet any of criteria for having a palette, so they were classified as “Without palette”. That is, most of the covers do not create color harmony impression.

The next step of investigation is finding the closest palette type, according to which the designer can correct the cover image to match it with the skin colors that are predominant for romance novels. Calculations carried out in accordance with the ratios (7)–(22) showed that between images designated as “Without palette”, 758 are close to an analog palette, 108 to a monochromatic palette, 17 to a split palette, 152 to a triad palette, and 399 to a tetradic palette.

Closeness to a palette is considered as exceeding the threshold values of difference in angles between the color vectors of the LAB space and the reference values of the angles that define the palettes. In other words, we conclude that an image is close to a palette when one of the conditions is met:(24)$$\Delta_{I_{k}Ana\log}=\left|D_{I_{k}}-\cos\frac{\pi }{6}\right|\leq \left(\frac{1}{2}\cos\frac{\pi }{6}\right)$$(25)$$\Delta_{I_{k}Monochrom}=\left|D_{I_{k}}-\cos0\right|\leq \left(\frac{1}{2}\cos\frac{\pi }{6}\right)$$(26)$$\Delta_{I_{k}Compl}=\left|D_{I_{k}}-\cos\pi \right|\leq \left(\frac{1}{2}\cos\frac{\pi }{6}\right)$$(27)$$\Delta_{I_{k}Split}=\left|D_{I_{k}}-\cos\frac{5\pi }{6}\right|\leq \left(\frac{1}{2}\cos\frac{\pi }{6}\right)$$(28)$$\Delta_{I_{k}Triad}=\left|D_{I_{k}}-\cos\frac{2\pi }{3}\right|\leq \left(\frac{1}{2}\cos\frac{\pi }{6}\right)$$(29)$$\Delta_{I_{k}Tetrad}=\left|D_{I_{k}}-\cos\frac{\pi }{4}\right|\leq \left(\frac{1}{2}\cos\frac{\pi }{6}\right)$$

To set the tolerance of angular deviation, a value $\left(\frac{1}{2}\cos\frac{\pi }{6}\right)$ was chosen as half of the smallest of the angles: π/6, 0, π, 2π/3, 5π/6, π/4.

So, the final conclusion is that for romantic novel covers, designers most often choose elements painted in skin colors and forming an analog color palette or a combination close to it. The complementary palette is almost never used for such publications; an alternative to the analog palette is a tetradic or triadic one.

### 4.2. Cover Design with Generative Services for Self-Publishing

An author who wants to design a book on their own often refuses the services of professionals for financial reasons. Publishing house services for creating a cover can cost from USD 300–700 up to USD 2000, freelance services cost -USD 15–100, and even in paid specialized generative services, one cover template costs USD 60–150 [47]. In addition, the author may believe that only they knows their book and their audience better than others; therefore, only they are able to express their understanding visually most accurately. That is why this study considered free image generation AI-based services: Dreamlike [48], Leonardo Ai [49], Leap [50], and Playground [51].

As object of research, the cover classified as “Without palette” (Figure 8) was chosen. In the original dataset [43], this cover is accompanied by the author’s name, title, and brief description: “Opens a contemporary western series set in Gold Valley, Ore. Olivia Logan has a plan: win back her ex by fake-dating someone else and making him jealous. Laid-back cowboy Luke Hollister wants her help convincing her father to sell him land, and agrees to help her if she’ll help him. Naturally, their plans go awry”. This description was accepted as a prompt for generative services in task cover correction or development.

The quality of the design created by GAI was visually assessed based on simple criteria:-Compliance of the color scheme with the proposed prompt-Compliance of the image meaning with the proposed prompt-Presence of phantom font-like elements in the images

It is noticeable that skin tones are present in the design, so it should be changed to an analog palette that matches the skin tones. This condition was added to the verbal prompt: “Analog color palette close to light-beige”. In addition, the cover features a photorealistic image of a man and a woman, so this option should also be used to generate the new variant.

Let us consider the tools of generative services that will be useful for solving the task—improving the existing version of the cover by changing the color palette. Settings common to all services studied, such as seed, image format, negative prompts, speed, quality, etc., are not key in this context, so they are not compared.

Dreamlike [48] proposes three main developing options: Generate/Edit/Enhance. Important possibility—uploading an initial image—is very useful for this research. Service allows choosing the illustration style by changing a model from the list: Kandinsky 2.0 (General), Neurogen 1.0 (General), Realism Engine 1.0 (Photoreal), Counterfeit 2.5 (Anime), Dreamlike Diffusion 1.0 (Artistic), Dreamlike Anime 1.0 (Anime), Dreamlike Photoreal 2.0 (Photorealistic), and Stable Diffusion 1.0 (General, base model). In this service, the prompt size is limited (no more than 56 words) for some models.

For our task, Kandinsky 2.0 (General), Neurogen 1.0 (General), and Dreamlike Photoreal 2.0 (Photorealistic) were chosen, but for these models, prompts bigger than 56 words are not acceptable; so, Stable Diffusion 1.0 (General, base model) also was investigated. Results for queries are shown in Figure 9.

Dreamlike Photoreal 2.0 is given the highest-quality results; the enhancement did not significantly change the design. In both cases, there remains a problem with generating the title and the author’s name since it will be difficult to remove their imitation from the picture without interference—they are on a gradient background. Obviously, the negative prompt “no titles” needs to be used.

Leonardo Ai [49] has some platform presets that help to choose a trend: Leonardo Phoenix, Anime, Cinematic Kino, Concept Art, Graphic Design, Illustrative Albedo, Leonardo Lighting, Lifelike Vision, Portrait Perfect, Stock Photography. Adding to this set there are some preset styles: 3D Render, Bokeh, Cinematic, Cinematic Concept, Creative, Dynamic, Fashion, Graphic Design Pop Art, Graphic Design Vector, Illustration, Macro, Minimalist, Moody, None, Portrait, Photography, Ray Traced, Sketch, Stock Photo, and Vibrant.

For our task, the Leonardo Phoenix platform turned out to be the most suitable. Styles were researched: Vibrant, Minimalist, Dynamic, and Graphic Design Vector. Results for queries are shown in Figure 10. The choice of Graphic Design Vector is explained by the fact that illustrations for printing are often prepared not in raster but in vector editors, which ensures higher image quality. In addition, [47] notes that there is a tendency to increase the number of hand-drawn rather than photorealistic covers.

It is obvious that all variants have high quality, which corresponds to text prompt and palette restriction. Some designs contain a mistaken title, but in vector images, it can be easy to correct.

Leap [50], at first glance, differs from its competitors by a small selection of settings for generation. The service provides the ability to enter a prompt up to 1500 characters, set the image format, and explore styles: Marketing, People, Photography, Art, Products, Landscapes, Logos and Icons, and Interior Design. Despite the apparent lack of alternatives when choosing styles, the service provided high-quality illustrations that fully met the query (Figure 11) without exploring.

Playground [51] provides advanced options for illustrations: options Create, Explore, and Change Colors. There is a large selection of design options for various types of products in the Create option: Logo, T-shirt, Social Media Post, Art, Poster, Mobile Wallpaper, Mock Ups, Stickers, Cards and Invites, Seamless Patterns, Memes, Monogram, EBook Cover, Virtual Backgrounds, Coloring Book Pages, and Mug.

The Change Colors option provides more than thirty palettes, which guarantees harmonious color content even without verbal prompts. The Explore option also adds style solutions. Results for queries are shown in Figure 12.

## 5. Discussion

The research on color usage in book cover design has been successfully completed, offering insights into genre-specific preferences and defining optimal color palettes. This research revealed that each genre benefits from distinct color schemes that align with reader expectations and psychological associations.

A review of generative services, such as AI-powered design platforms, demonstrated their potential to enhance the book cover creation process. These tools enable rapid prototyping by analyzing input data like keywords, genres, and thematic elements. They suggest color palettes and visual arrangements that resonate with market trends and audience preferences, significantly reducing the time required for ideation and iteration.

But, on the other hand, AI in book cover design can sometimes produce generic or unoriginal results, as it heavily relies on existing training data for large language models used in GAI. It also struggles with understanding complex cultural nuances or emotional subtleties, often leading to designs that may not resonate with the target audience. While GAI can quickly generate many versions, it can miss the finer details and context of a book’s themes, potentially missing the true essence of the book. Serious drawbacks inherent in existing AI systems in design are the devaluation of human creativity and the risk of plagiarism, as AI can reproduce copyrighted or culturally insensitive content.

The proposed operation sequence for developing a book cover design integrates positive insights and progressive technologies, creating a streamlined workflow. All studied examples show the same drawback inherent in other generative services researched—the presence of unrealistic inscriptions, which correction requires additional manual editing of the illustration. Thus, the study of the possibility of using generative services for developing book cover designs demonstrates the availability and broad functionality of such services. However, their use for this purpose will not be a one-step process. None of the generative services studied provide the ability to enter a short description of a book in one iteration and immediately receive a finished design, which is an ideal solution for the self-publishing process. Therefore, it becomes necessary to generalize the sequence of actions when developing a book cover using generative artificial intelligence (GAI). The sequence diagram can be seen in Figure 13.

The process begins with defining the book’s genre, theme, and target audience. Color clustering based on machine learning shows most frequently used colors for a genre (if appropriate data are available). Based on this information, generative services can suggest initial layouts. The designer develops a prompt as a query for GAI tools. The designer (who is also considered the book author in self-publishing) launches the stages of Exploring styles and constraints, forming and initializing conditions and constraints for generation (Conditioning), then the GAI service forms a response to request and the designer selects the most suitable variants, combining them with additional conditions if necessary. Here, in the work process, small pauses (Intermediate Stops) are possible, associated with the need to comprehend the results obtained and make a decision on their compliance with the design goal set. Then, if additional conditions are present, the generative service is relaunched (Evolving), the designer selects suitable results, and the process ends. Designers then refine the outputs, incorporating specific imagery, typography, and branding elements to ensure alignment with the book’s tone and narrative.

After generating prototypes, the next step involves testing the cover’s impact. Feedback from focus groups or market-specific algorithms helps validate the design’s effectiveness. Adjustments are made based on this feedback, focusing on color harmony, legibility, and emotional resonance. The finalized design undergoes format optimization to ensure suitability for both print and digital platforms, including thumbnails for e-commerce sites.

It is clear that such a sequence of actions is slightly more complex compared with a one-step procedure that is desirable for self-publishing; nevertheless, it is almost linear, and it can be recommended for cases of working with any generative service when creating a book cover design.

In this paper, it is proposed to use the k-means clustering method, which is simple and highly efficient. The fact that the algorithm requires a pre-set number of clusters in this case is another advantage, since this parameter can be used to adjust the level of accuracy and complexity of the task—‚the greater the number of clusters, the more difficult it is to determine to which color palette the image content will be assigned. Another aspect that determines the simplicity of the proposed approach is the use of GAI to correct the color content of images. As can be seen from the results of the experiment, each of the generative services considered not only changes the color scheme of the cover but also completely changes the entire image while remaining within the genre.

Thus, it can be expected that after the interaction of the designer with generative service in accordance with the developed scheme of Figure 13, the service will not only perform color correction but also offer a design that more fully meets the content of the book.

## 6. Conclusions

This research and proposed process underscore the importance of integrating creative expertise with modern tools. By combining data-driven insights about genre-specific color preferences with the efficiency of generative services, designers can produce book covers that are both visually compelling and commercially successful. Designers who create book cover layouts can use the proposed methodology to study consumer preferences in a particular genre, evaluate the conformity of the created layout version with these preferences, and step-by-step modify the layout to harmonize the color content. The approach ensures that covers resonate with readers, align with market trends, and support the book’s broader marketing strategy.

Future research in this area may involve improving generative artificial intelligence systems that will allow new versions of image design layouts to be proposed by only partially modifying existing versions and making more subtle changes to them, for example, by performing fine gradation correction of the image taking into account its meaning and its elements.

The use of artificial intelligence in book cover design is a promising direction. One of the main advantages of such systems is the ability to generalize huge amounts of information—both text and visual. Its combination with algorithms for fine-tuning readers’ preferences depending on age category, genre aesthetics, and target audience’s ethnic characteristics opens up broad opportunities for greater individualization of book production and further development of artificial intelligence systems.

## Figures and Tables

**Figure 1 jimaging-11-00046-f001:**
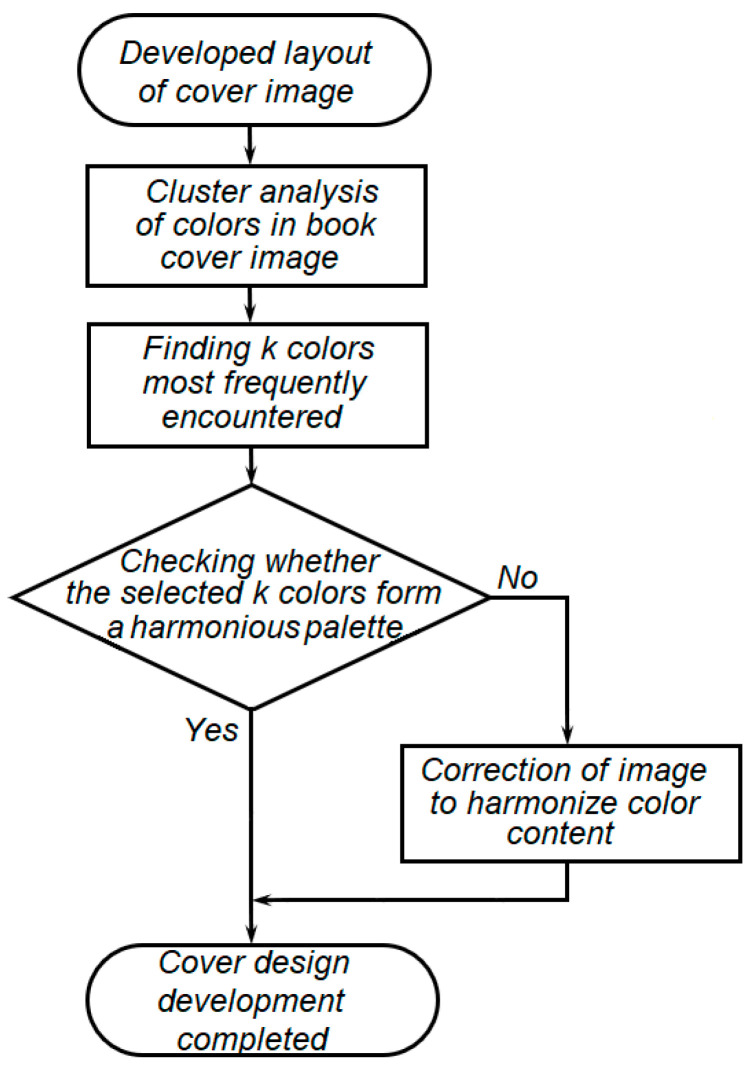
The sequence of research tasks solving.

**Figure 2 jimaging-11-00046-f002:**
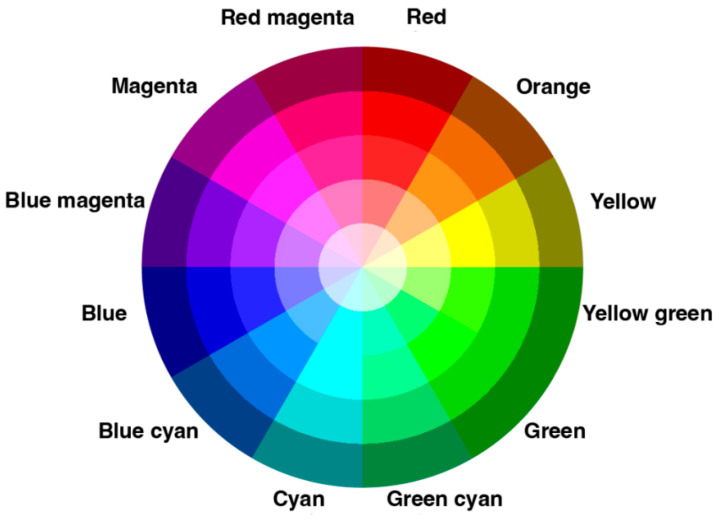
RGB color wheel by RMIT, licensed under CC BY-NC 4.0 [36].

**Figure 3 jimaging-11-00046-f003:**
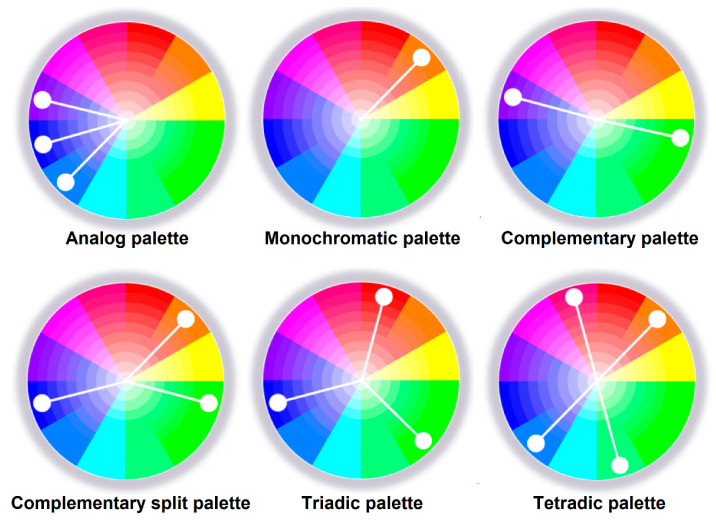
Types of color palettes [36].

**Figure 4 jimaging-11-00046-f004:**
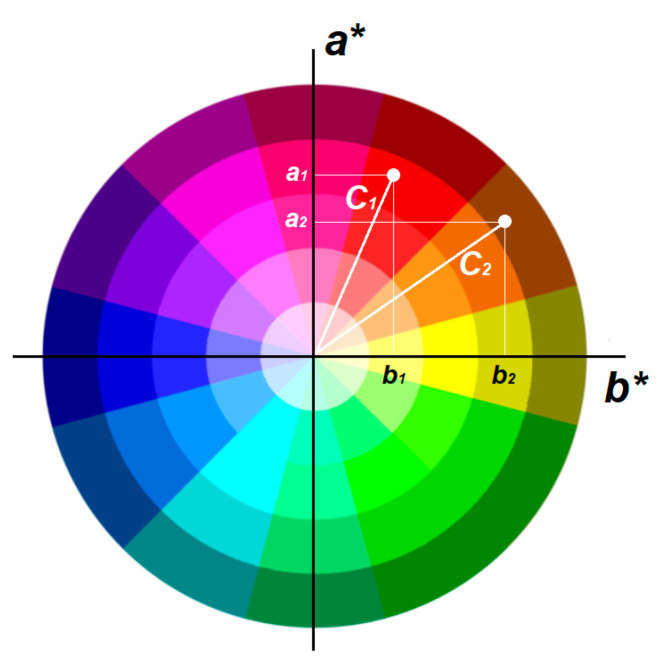
Radius vectors to two colors $C_{1}$ and $C_{2}$ from wheel center.

**Figure 5 jimaging-11-00046-f005:**
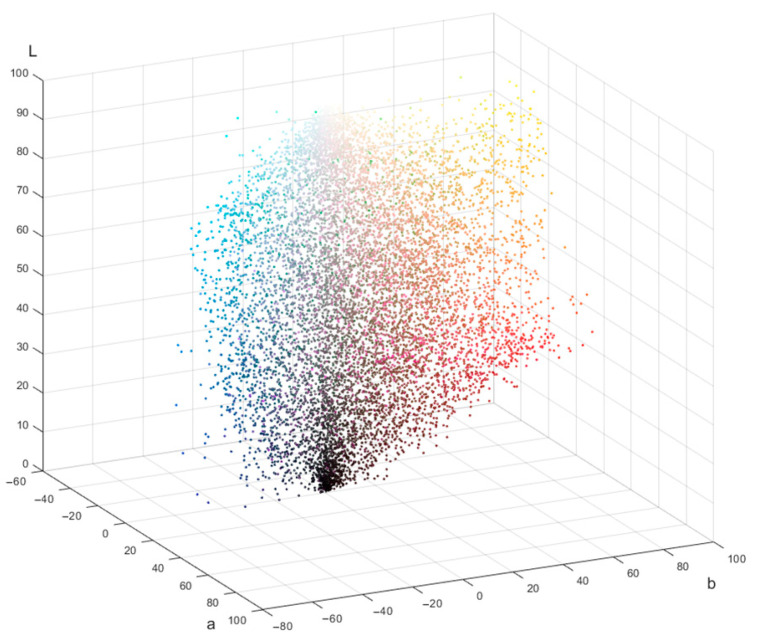
The 10,038 colors in CIE L*a*b* space that were identified by k-means clustering on book covers from dataset [43].

**Figure 6 jimaging-11-00046-f006:**
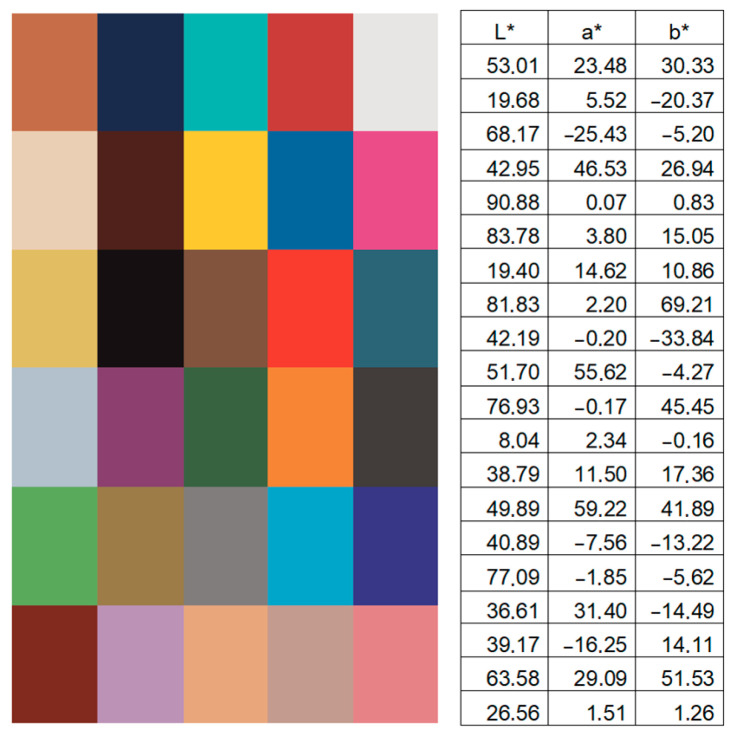
The 30 most frequently repeated colors on covers from dataset [43], and their CIE L*a*b* coordinates.

**Figure 7 jimaging-11-00046-f007:**
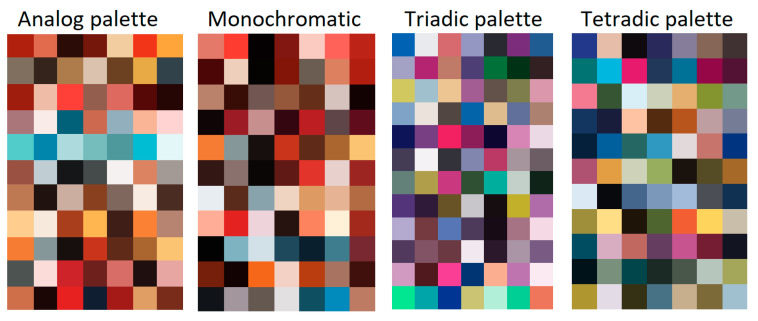
Examples of 7 most frequently occurring colors on each cover that satisfy condition (22) and form analog, monochromatic, triadic, and tetradic palettes.

**Figure 8 jimaging-11-00046-f008:**
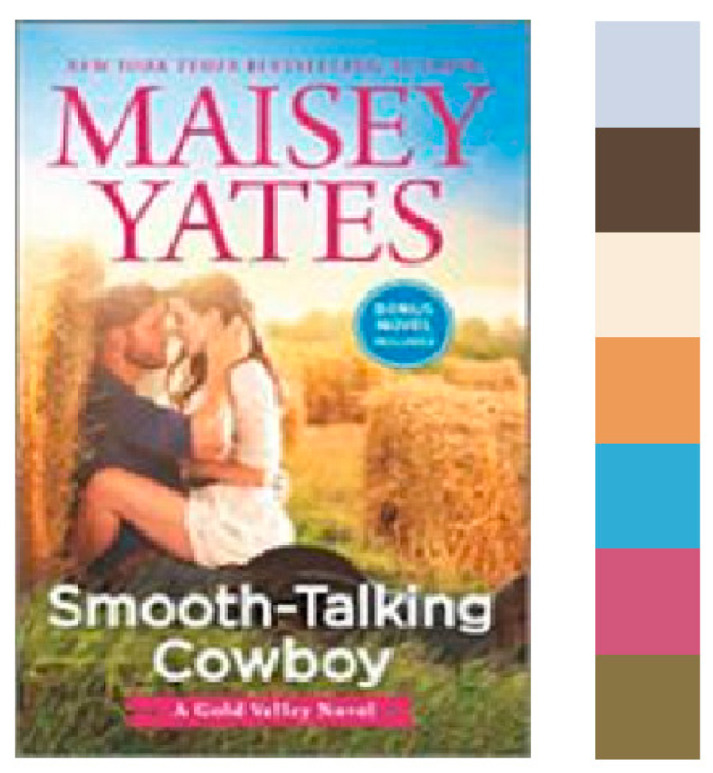
The cover classified as “Without palette” and seven most frequent colors identified using k-means clustering.

**Figure 9 jimaging-11-00046-f009:**
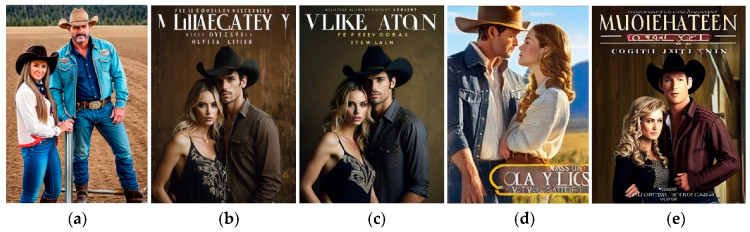
Dreamlike results for romance cover using models (**a**) Kandinsky 2.0 (General); (**b**) Dreamlike Photoreal 2.0 (Photorealistic); (**c**) Dreamlike Photoreal 2.0 (Photorealistic) with enhancing; (**d**) Neurogen 1.0 (General); and (**e**) Stable Diffusion 1.0 (General, base model).

**Figure 10 jimaging-11-00046-f010:**
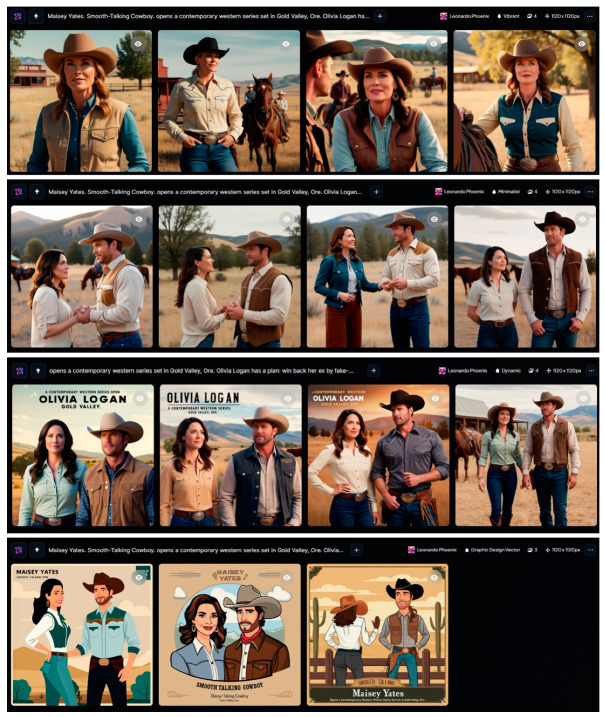
Leonardo Ai results for romance cover using Leonardo Phoenix platform with styles (in rows from **top** to **bottom**): Vibrant, Minimalist, Dynamic, and Graphic Design Vector.

**Figure 11 jimaging-11-00046-f011:**
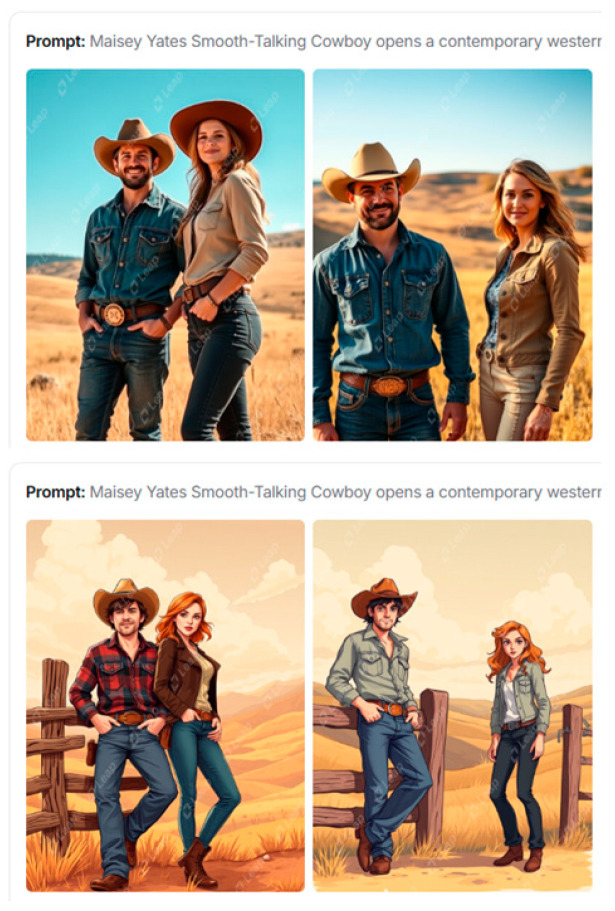
Leap results for romance cover.

**Figure 12 jimaging-11-00046-f012:**
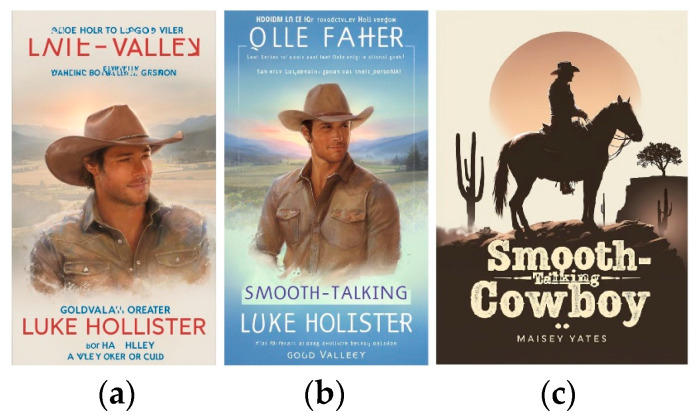
Playground results for romance cover with option Create/EBook Cover: (**a**) original prompt color palette; (**b**) PeacefulWaters palette of Change Colors; (**c**) silhouetted design after Explore option.

**Figure 13 jimaging-11-00046-f013:**
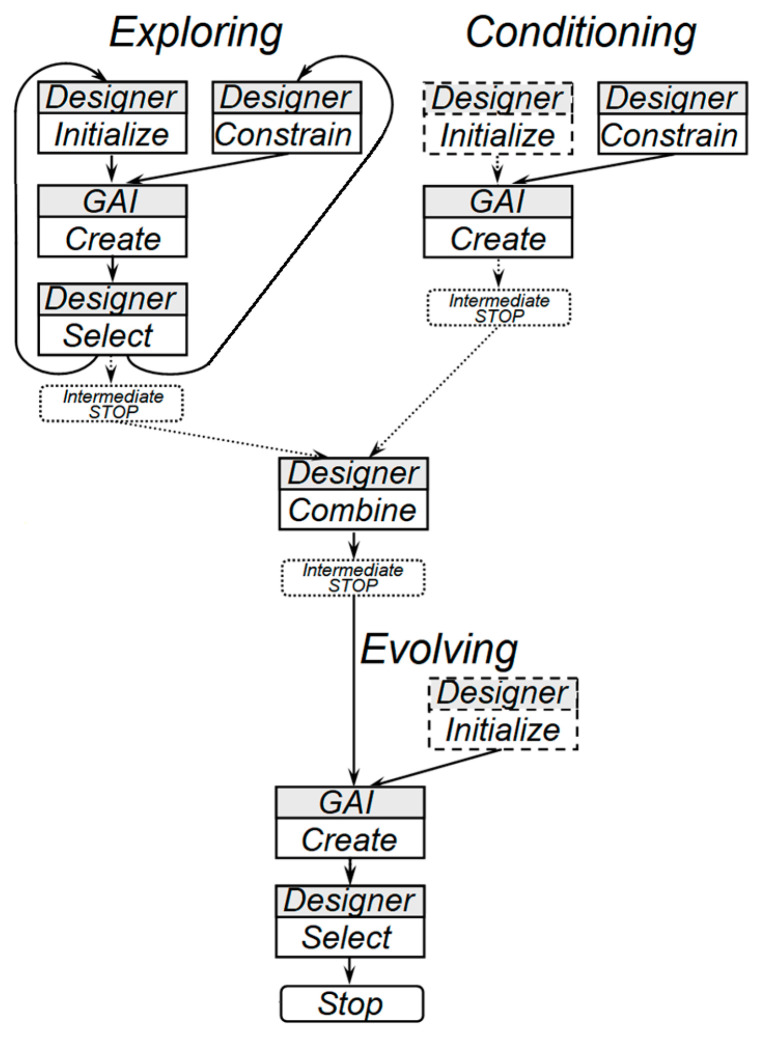
The diagram of operations sequence using GAI in creating a book cover design in the self-publishing process.

## Data Availability

The original contributions presented in this study are included in the article; further inquiries can be directed to the corresponding author.

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
