# Peer review of "Using Machine Learning and Generative Intelligence in Book Cover Development"

_2313-433X, 2025, doi:10.3390/jimaging11020046_

Round 1

Reviewer 1 Report

Comments and Suggestions for Authors

Abstract

Line 12–13: The claim regarding the identification of color preferences inherent in specific genres is intriguing. However, the term “color preferences” requires clarification—whether it refers to reader perception, market trends, or artistic design choices (no more than 2 sentences).

Line 15–16: The flowchart for using generative AI in book cover design makes a valuable contribution, but the abstract should expand its description to highlight specific advantages over traditional workflows (1 clause or 1 sentence).

Introduction

Line 24–26: The statistics on the global book market provide useful context but lack references to recent publications or databases to support these numbers.

Line 30–35: The increasing importance of book cover design is well-articulated, but it would benefit from examples of recent industry trends or high-profile successes attributed to innovative cover designs.

Line 93–94: The definition of a book cover as both an art form and a marketing tool is accurate but lacks references to research on consumer behavior or visual psychology in design.

Line 117–132: The list of current design trends is comprehensive. However, consider elaborating on how these trends have been influenced by cultural shifts or technological advancements, particularly AI.

Methodology

Line 150–151: The use of k-means clustering for color analysis is appropriate, but the rationale for selecting this algorithm over other clustering methods, such as DBSCAN or hierarchical clustering, is not discussed, at least 2 or 3 sentences with some references.

Line 165–168: The adoption of the CIE Lab* color space is justified for its perceptual accuracy. However, a brief comparison with alternative color spaces (e.g., RGB, CMYK) would strengthen this section.

Results

Line 285–287: The observation that skin tones dominate romantic novel covers is insightful but would benefit from references to reader expectations or market research supporting this trend.

Line 304–305: The classification of “Without palette” covers into closest palettes is valuable. However, the methodology for determining “closeness” could be elaborated further, especially regarding the thresholds used.

Discussion

Line 402–405: The effectiveness of generative services for prototyping is well-discussed. However, the drawbacks (e.g., limited control over output or ethical concerns with AI) need further emphasis.

Line 439–440: The recommendation to optimize covers for both print and digital formats is practical. Including examples of specific optimizations for thumbnails could improve this section.

Conclusion: The conclusion emphasizes the integration of data-driven insights and generative tools effectively. However, specific recommendations for future work, such as refining AI capabilities to include genre-specific aesthetics, would be valuable.

Author Response

Dear Reviewer, thank you for your valuable comments. The authors have done their best to correct them and hope that the article is now better. Here we present the changes that will be made to the text of the article after your comments. In maniscript they were marked cyan.

Abstract

Line 12–13: The claim regarding the identification of color preferences inherent in specific genres is intriguing. However, the term “color preferences” requires clarification—whether it refers to reader perception, market trends, or artistic design choices (no more than 2 sentences).

When choosing a book, readers often have certain expectations regarding the design of the publication, including the color of the cover. These expectations can be called color preferences, and they can depend on the genre of the book, its target audience, and even personal associations. Cultural context can also influence color choice, as certain colors can symbolize different emotions or moods in different cultures.

Line 15–16: The flowchart for using generative AI in book cover design makes a valuable contribution, but the abstract should expand its description to highlight specific advantages over traditional workflows (1 clause or 1 sentence).

An improved flow chart for using GAI in creating book covers in the process of self-publishing is proposed, which includes new stages, namely: exploring, conditioning and evolving. At these stages, designer creates prompts for GAI, examines how they and GAI’s issuances correspond to the task. Conditioning allows for even more precise adjustment of prompts to features of each book, and the evolving stage also includes post-processing of results already received from GAI. Post-processing, in turn, can be performed both in generative services and by a designer.

Introduction

Line 24–26: The statistics on the global book market provide useful context but lack references to recent publications or databases to support these numbers.

The global books market was valued at approximately USD 144.67 billion in 2023, and it is expected to grow at a compound annual growth rate, according to various estimates, from 1.8 to 3% from 2024 to 2030 [1 - 3].

Line 30–35: The increasing importance of book cover design is well-articulated, but it would benefit from examples of recent industry trends or high-profile successes attributed to innovative cover designs.

Nowadays, Generative Artificial Intelligence (GAI) tools are actively included in the design creation process. A 2024 report from Design Intelligence reveals that 60% of publishers have incorporated AI tools into various stages of book production, with 35% using them specifically for cover design. In addition, a survey by BookTech showed that 70% of self-published authors experimented with AI-powered design tools in 2023, a significant increase from 45% in 2020 [7].

Line 93–94: The definition of a book cover as both an art form and a marketing tool is accurate but lacks references to research on consumer behavior or visual psychology in design.

A book cover is both an art form and a marketing tool, requiring careful consideration of visual and psychological factors [12, 13].

Line 117–132: The list of current design trends is comprehensive. However, consider elaborating on how these trends have been influenced by cultural shifts or technological advancements, particularly AI.

The rise of AI in book cover design has significantly transformed the creative process, allowing designers to generate and experiment with endless visual concepts quickly and efficiently, thus opening doors to more experimental and diverse styles. Nostalgic trends referencing the 50s, 60s, and 70s often tap into cultural cycles of revival, where AI tools can quickly emulate and remix past design aesthetics, connecting contemporary readers with vintage visual cues. The increasing popularity of abstract art and mixed digital and hand-drawn styles reflects a desire for both innovation and authenticity, where AI allows seamless blending of different techniques and a more accessible way to merge traditional and modern forms. Minimalism with a twist and activism-themed designs are shaped by global social movements and the digital age, where simplicity and bold statements can be emphasized through AI-generated variations or optimized visual messaging that resonates with modern sensibilities. Trends like collages, double exposure, and movie-inspired covers benefit from the flexibility of AI, which can manipulate images and fonts in innovative ways, helping to tell visually complex stories that align with the growing need for immersive, visually dynamic book marketing.

Methodology

Line 150–151: The use of k-means clustering for color analysis is appropriate, but the rationale for selecting this algorithm over other clustering methods, such as DBSCAN or hierarchical clustering, is not discussed, at least 2 or 3 sentences with some references.

K-means is often considered better than DBSCAN and hierarchical clustering for large datasets because it is computationally more efficient, with a time complexity of O(nk), where n is the number of data points and k is the number of clusters. Hierarchical clustering forms a multi-level data architecture that can be scaled. K-means is limited to one level of data partitioning, typically performs better when the clusters are roughly spherical and evenly sized,  so it is much more effective in analyzing big data [28 - 30, 32]. DBSCAN takes into account outliers in the data, grouping them into a separate cluster. This can be useful for data outliers detecting. But when it is unknown whether there will be outliers at all in the task, when generalized information about presence of colors in a large array of images is more important, such features of DBSCAN seem redundant. In addition, DBSCAN is focused on forming clusters with uniform density, but with regard to color data from images, it cannot be guaranteed that they will actually fill the color space uniformly, rather the opposite.

Line 165–168: The adoption of the CIE Lab* color space is justified for its perceptual accuracy. However, a brief comparison with alternative color spaces (e.g., RGB, CMYK) would strengthen this section.

CIE L*a*b* is device-independent color space, as opposed to, for example, RGB, and CMYK. RGB (Red, Green, Blue) is an additive model of device-dependent color space, commonly used for digital displays, where colors are created by combining varying intensities of red, green, and blue light [31]. CMYK (Cyan, Magenta, Yellow, Key/Black) is a subtractive color model used in color printing, where colors are produced by subtracting light using ink, and it works by mixing various percentages of cyan, magenta, yellow, and black inks [31]. This color space is device-dependent too. The main difference between these three color spaces lies in their purpose for consistent color reproduction across different devices and environments.

Results

Line 285–287: The observation that skin tones dominate romantic novel covers is insightful but would benefit from references to reader expectations or market research supporting this trend.

Despite the fact that in recent years, cover designs with images of fully or partially naked people have become less and less popular, this study found, that among the 30 most common colors on romance covers, 9 (that is, almost a third) can be classified as skin tones. These colors were compared with data from [44, 45]. 6 colors are shades of blue, 5 are red and violet, 4 are shades of neutral gray, 4 are shades of yellow-brown, and only 2 colors are green. So, the discovered now fact is that a third of the most common colors on romance covers are skin colors, despite the images of partially clothed people are becoming fewer. It is clear from the clustering results that skin colors gamut in the perception of designers, authors and readers is closely related to romantic novels content.

Line 304–305: The classification of “Without palette” covers into closest palettes is valuable. However, the methodology for determining “closeness” could be elaborated further, especially regarding the thresholds used.

To decide if color correction an image  is necessary, we need to determine to what palette does it belong to.

Calculations carried out in accordance with the ratios (7) – (22) showed that between images designated as "Without palette" 758 are close to Analog palette

Closeness to palette is considered as exceeding the threshold values ​​of difference in angles between the color vectors of the LAB space and the reference values ​​of the angles that define the palettes. In other words, we conclude, that image is close to palette, when one of the conditions is met:

,                                                                (23)

 ,                                                          (24)

   ,                                                (25)

   ,                                                 (26)

,                                                                (27)

.                                                              (28)

To set the tolerance of angular deviation, a value  was chosen as half of the smallest of the angles: , 0, , , , .

Discussion

Line 402–405: The effectiveness of generative services for prototyping is well-discussed. However, the drawbacks (e.g., limited control over output or ethical concerns with AI) need further emphasis.

But, on the other hand, AI in book cover design can sometimes produce generic or unoriginal results, as it relies heavily on existing training data for large languadge models used in GAI. It also struggles with understanding complex cultural nuances or emotional subtleties, often leading to designs that may not resonate with the target audience. While GAI can quickly generate many versions, it can miss the finer details and context of a book’s themes, potentially missing the true essence of the book. Serious drawbacks inherent in existing AI systems in design are the devaluation of human creativity and the risk of plagiarism, as AI can reproduce copyrighted or culturally insensitive content.

Line 439–440: The recommendation to optimize covers for both print and digital formats is practical. Including examples of specific optimizations for thumbnails could improve this section.

Conclusion: The conclusion emphasizes the integration of data-driven insights and generative tools effectively. However, specific recommendations for future work, such as refining AI capabilities to include genre-specific aesthetics, would be valuable.

The use of artificial intelligence in book cover design is a promising direction. One of the main advantages of such systems is the ability to generalize huge amounts of information - both text and visual. Its combination with algorithms for fine-tuning readers' preferences depending on age category, genre aesthetics, and target audience ethnic characteristics opens up broad opportunities for greater individualization of book production and further development of artificial intelligence systems.

Reviewer 2 Report

Comments and Suggestions for Authors
  1. The paper presents a fascinating and innovative approach to improving book cover design using machine learning and GAI. The integration of color clustering and design principles into the self-publishing process is a timely and valuable contribution to the field. The authors effectively highlight the evolution of book cover design, especially with the rise of AI tools. The historical background adds context to the significance of book covers in both traditional and digital publishing. The methodology section is thorough, detailing the application of the k-means clustering method and its relevance to identifying color trends in book covers. This provides a solid basis for the research and enhances the paper's credibility. The discussion on AI-powered generative services is well-rounded, offering insights into how these tools can assist self-published authors. The detailed exploration of specific platforms like Dreamlike, Leonardo Ai, and Playground adds practical value to the research. The introduction provides a strong foundation for the paper's argument. However, it might benefit from more direct references to current AI technologies and their specific impacts on the publishing industry. This could help better situate the reader in the context of the rapid technological advancements in the field. The methodology involving k-means clustering to analyze color palettes in book covers is clear. However, the authors could expand on how the selected color clusters align with psychological and cultural associations within book genres. This would further enhance the practical application of the research findings. The results, especially the analysis of romance book covers, are compelling. It would be beneficial to include more visual examples of the clustered data or a graphical representation of how the color schemes map to specific genres. This would aid in better illustrating the research findings. The discussion on generative services is insightful. However, there is room for further elaboration on the challenges and limitations of AI-generated book covers. For example, how might the integration of human creativity and AI-driven designs influence overall market success in terms of sales and reader engagement? The conclusion successfully ties together the findings and emphasizes the importance of combining creative expertise with AI tools. It would be helpful to include some recommendations for authors or designers who want to integrate AI into their cover design process, based on the study's results. 

Please consider the following paper in the related work section:
Elastic deep multi-view autoencoder with diversity embedding

Author Response

Dear Reviewer, thank you for your valuable comments. The authors have done their best to correct them and hope that the article has now become better.
Here we present the changes that will be made to the text of the article after your comments.

However, it might benefit from more direct references to current AI technologies and their specific impacts on the publishing industry. This could help better situate the reader in the context of the rapid technological advancements in the field.

Nowadays, Generative Artificial Intelligence (GAI) tools are actively included in the design creation process. A 2024 report from Design Intelligence reveals that 60% of publishers have incorporated AI tools into various stages of book production, with 35% using them specifically for cover design. In addition, a survey by BookTech showed that 70% of self-published authors experimented with AI-powered design tools in 2023, a significant increase from 45% in 2020 [7].

The methodology involving k-means clustering to analyze color palettes in book covers is clear. However, the authors could expand on how the selected color clusters align with psychological and cultural associations within book genres. This would further enhance the practical application of the research findings.

According to [39], red colors are associated with love, purple and pink - with beauty and femininity; blue shades - with purity. In romance books, for example, love often emerges as a transformative force, bringing characters closer and allowing them to discover deeper connections. Beauty, femininity, and purity are frequently depicted through idealized characters or settings, where these traits symbolize innocence, emotional depth, and the potential for a perfect love story.

The results, especially the analysis of romance book covers, are compelling. It would be beneficial to include more visual examples of the clustered data or a graphical representation of how the color schemes map to specific genres. This would aid in better illustrating the research findings.

Figure 7. Examples of 7 most frequently occurring colors on each cover, that satisfy condition (22) and form Analog, Monochromatic, Triadic and Tetradic palettes

The discussion on generative services is insightful. However, there is room for further elaboration on the challenges and limitations of AI-generated book covers. For example, how might the integration of human creativity and AI-driven designs influence overall market success in terms of sales and reader engagement?

Nowadays, Generative Artificial Intelligence (GAI) tools are actively included in the design creation process. A 2024 report from Design Intelligence reveals that 60% of publishers have incorporated AI tools into various stages of book production, with 35% using them specifically for cover design. In addition, a survey by BookTech showed that 70% of self-published authors experimented with AI-powered design tools in 2023, a significant increase from 45% in 2020 [7]

The conclusion successfully ties together the findings and emphasizes the importance of combining creative expertise with AI tools. It would be helpful to include some recommendations for authors or designers who want to integrate AI into their cover design process, based on the study's results. 

Designers who create book cover layouts can use the proposed methodology to study consumer preferences in a particular genre, evaluate the conformity of the created layout version with these preferences, and step-by-step modify the layout to harmonize the color content. The approach ensures covers resonate with readers, align with market trends, and support the book's broader marketing strategy.

The use of artificial intelligence in book cover design is a promising direction. One of the main advantages of such systems is the ability to generalize huge amounts of information - both text and visual. Its combination with algorithms for fine-tuning readers' preferences depending on age category, genre aesthetics, and target audience ethnic characteristics opens up broad opportunities for greater individualization of book production and further development of artificial intelligence systems

Тo determine whether there are established semantic links between book genres, design styles and color palettes of published book covers, various methods of machine learning and artificial intelligence are used [28 - 30]

Reviewer 3 Report

Comments and Suggestions for Authors

Author proposed NLP based method for book cover development. The highlight of this work is that it uses clustering method to group the preferences of the user and based on this the suggestion will be given while developing book covers. Despite this novelty, several drawbacks persist. 

1. Experimental results obtained should be mentioned in the abstract.

2. Research background and the problem statement is not clear.

3. Related works needs to be enhanced. Author should study several works to improve the quality of the manuscript.

4. An introduction to GenAI and its applications can be highlighted. This will improve the readability.

5. The methodology should be explained with neat diagram or an algorithmic approach.

6. The complexity analysis can be done for the proposed work.

7. Dataset utilized for the analysis can be discussed.

8. How author assess the performance of the GenAI model for generating the book covers?

9. The limitations and the future work can be discussed.

Comments on the Quality of English Language

Author proposed NLP based method for book cover development. The highlight of this work is that it uses clustering method to group the preferences of the user and based on this the suggestion will be given while developing book covers. Despite this novelty, several drawbacks persist. 

1. Experimental results obtained should be mentioned in the abstract.

2. Research background and the problem statement is not clear.

3. Related works needs to be enhanced. Author should study several works to improve the quality of the manuscript.

4. An introduction to GenAI and its applications can be highlighted. This will improve the readability.

5. The methodology should be explained with neat diagram or an algorithmic approach.

6. The complexity analysis can be done for the proposed work.

7. Dataset utilized for the analysis can be discussed.

8. How author assess the performance of the GenAI model for generating the book covers?

9. The limitations and the future work can be discussed.

Author Response

Dear Reviewer, thank you for your valuable comments. The authors have done their best to correct them and hope that the article has now become better.
Here we present the changes that will be made to the text of the article after your comments.

  1. Experimental results obtained should be mentioned in the abstract.

The experiment allowed us to use machine learning method to determine which colors are most often found in the design of book covers of one of the genres, and to check whether these colors correspond to harmonious color palettes. In accordance with the proposed scheme of the design process using generative artificial intelligence, versions of book cover layouts of a given genre were obtained.

  1. Research background and the problem statement is not clear.

The purpose of this study is to improve the quality of book cover design by creating a harmonious color content of images. To achieve this goal, it is necessary to:

- develop a methodology for determining which colors predominate in the developed cover design layout;

- assess whether the color scheme of the design is harmonious;

- if the colors selected for the cover layout are not harmonious, study the capabilities of generative services based on artificial intelligence for correcting and harmonizing colors in the layout.

  1. Related works needs to be enhanced. Author should study several works to improve the quality of the manuscript.

Тo determine whether there are established semantic links between book genres, design styles and color palettes of published book covers, various methods of machine learning and artificial intelligence are used [28 - 30]. In this paper, it is proposed to use a well-known image clustering method k-means [31].

K-means is often considered better than DBSCAN and hierarchical clustering for large datasets because it is computationally more efficient, with a time complexity of O(nk), where n is the number of data points and k is the number of clusters. Hierarchical clustering forms a multi-level data architecture that can be scaled. K-means is limited to one level of data partitioning, typically performs better when the clusters are roughly spherical and evenly sized, so it is much more effective in analyzing big data [28 - 30, 32]. DBSCAN takes into account outliers in the data, grouping them into a separate cluster. This can be useful for data outliers detecting. But when it is unknown whether there will be outliers at all in the task, when generalized information about presence of colors in a large array of images is more important, such features of DBSCAN seem redundant. In addition, DBSCAN is focused on forming clusters with uniform density, but with regard to color data from images, it cannot be guaranteed that they will actually fill the color space uniformly, rather the opposite.

  1. An introduction to GenAI and its applications can be highlighted. This will improve the readability.

The swift progress of technologies leveraging artificial intelligence, particularly text-to-image generation, is transforming our approach to design across various fields. This includes game design [37 - 39], web design [40], and multimedia or print publications [41]. As a result of numerous similar studies, a distinct field known as generative artificial intelligence (GAI) has emerged. This method is undeniably appealing to professionals due to its virtually limitless creative potential, offering numerous options and speeding up the process of producing artistic concepts.

Generative Adversarial Networks (GANs) [42] were among the first solutions for the artificial creation of images containing objects, and have now become widely popular in image generation. GANs consist of two main components: the generator and the discriminator. The generator learns to replicate the statistical distribution of real examples to generate new data, while the discriminator attempts to distinguish whether the input data is real or fake. The design of both the generator and the discriminator plays a critical role in the stability and effectiveness of GANs. Traditional convolutional GANs generate high-resolution details by relying solely on spatially localized points in lower-resolution feature maps, which makes generating diverse high-resolution samples from complex datasets a challenging task.

  1. The methodology should be explained with neat diagram or an algorithmic approach.

The purpose of this study is to improve the quality of book cover design by creating a harmonious color content of images. To achieve this goal, it is necessary to:

- develop a methodology for determining which colors predominate in the developed cover design layout;

- assess whether the color scheme of the design is harmonious;

- if the colors selected for the cover layout are not harmonious, study the capabilities of generative services based on artificial intelligence for correcting and harmonizing colors in the layout.

The sequence of solving these problems is shown in the Figure 1.

Figure 1. The sequence of research tasks solving.

  1. The complexity analysis can be done for the proposed work.

In this paper, it is proposed to use the k-means clustering method, which is simple and highly efficient. The fact that algorithm requires a pre-set number of clusters in this case is another advantage, since this parameter can be used to adjust the level of accuracy and complexity of the task - the greater number of clusters, the more difficult it is to determine which color palette the image content will be assigned to. Another aspect that determines simplicity of proposed approach is use of GAI to correct the color content of images. As can be seen from the results of the experiment, each of the generative services considered not only changes the color scheme of the cover, but completely changes the entire image, while remaining within the genre.

Thus, it can be expected that after the interaction of designer with generative service in accordance with developed scheme of Figure 13, the service will not only perform color correction, but will offer a design that more fully meets the content of the book.

  1. Dataset utilized for the analysis can be discussed.

To explore the possibility of identifying the most commonly used color palettes for books of a certain genre, a dataset with romance book covers [43] was used. This dataset was gathered within twelve years 2011-2023 and contains 1434 cover images, author names and titles, brief book annotations. The images are photographs of book covers sold on various online services in the romance novel genre.

  1. How author assess the performance of the GenAI model for generating the book covers?

The quality of the design created by GAI was assessed visually based on simple criteria:

- compliance of the color scheme with the proposed prompt;

- compliance of the image meaning with the proposed prompt;

- presence of phantom font-like elements in the images.

  1. The limitations and the future work can be discussed.

 But, on the other hand, AI in book cover design can sometimes produce generic or unoriginal results, as it relies heavily on existing training data for large languadge models used in GAI. It also struggles with understanding complex cultural nuances or emotional subtleties, often leading to designs that may not resonate with the target audience. While GAI can quickly generate many versions, it can miss the finer details and context of a book’s themes, potentially missing the true essence of the book. Serious drawbacks inherent in existing AI systems in design are the devaluation of human creativity and the risk of plagiarism, as AI can reproduce copyrighted or culturally insensitive content.

Future research in this area may involve improving generative artificial intelligence systems that will allow new versions of image design layouts to be proposed by only partially modifying existing versions, making more subtle changes to them, for example, by performing fine gradation correction of the image taking into account its meaning and its elements.

Round 2

Reviewer 2 Report

Comments and Suggestions for Authors

All the mentioned comments have been applied completely.

Reviewer 3 Report

Comments and Suggestions for Authors

All the corrections were done. Hence, the paper can be considered for publication.

Comments on the Quality of English Language

Need to improve